# Multimodal measurement approach to identify individuals with mild cognitive impairment: study protocol for a cross-sectional trial

Bernhard Grässler ![ORCID],[1] Fabian Herold,[2] Milos Dordevic,[2] Tariq Ali Gujar,[1] Sabine Darius,[3] Irina Böckelmann,[3] Notger G Müller,[2,4] Anita Hökelmann[1]

¹Institute of Sport Science, Faculty of Humanities, Otto von Guericke University Magdeburg, Magdeburg, Germany
²Department of Neuroprotection, German Centre for Neurodegenerative Diseases Site Magdeburg, Magdeburg, Germany
³Occupational Medicine, Otto von Guericke University Medical Faculty, Magdeburg, Germany
⁴Department of Neurology, Otto von Guericke University Medical Faculty, Magdeburg, Germany

**Correspondence to**
Dr Bernhard Grässler;
bernhard.graessler@ovgu.de

## ABSTRACT

**Introduction** The diagnosis of mild cognitive impairment (MCI), that is, the transitory phase between normal age-related cognitive decline and dementia, remains a challenging task. It was observed that a multimodal approach (simultaneous analysis of several complementary modalities) can improve the classification accuracy. We will combine three noninvasive measurement modalities: functional near-infrared spectroscopy (fNIRS), electroencephalography and heart rate variability via ECG. Our aim is to explore neurophysiological correlates of cognitive performance and whether our multimodal approach can aid in early identification of individuals with MCI.

**Methods and analysis** This study will be a cross-sectional with patients with MCI and healthy controls (HC). The neurophysiological signals will be measured during rest and while performing cognitive tasks: (1) Stroop, (2) N-back and (3) verbal fluency test (VFT). Main aims of statistical analysis are to (1) determine the differences in neurophysiological responses of HC and MCI, (2) investigate relationships between measures of cognitive performance and neurophysiological responses and (3) investigate whether the classification accuracy can be improved by using our multimodal approach. To meet these targets, statistical analysis will include machine learning approaches.

This is, to the best of our knowledge, the first study that applies simultaneously these three modalities in MCI and HC. We hypothesise that the multimodal approach improves the classification accuracy between HC and MCI as compared with a unimodal approach. If our hypothesis is verified, this study paves the way for additional research on multimodal approaches for dementia research and fosters the exploration of new biomarkers for an early detection of nonphysiological age-related cognitive decline.

**Ethics and dissemination** Ethics approval was obtained from the local Ethics Committee (reference: 83/19). Data will be shared with the scientific community no more than 1 year following completion of study and data assembly.

**Trial registration number** ClinicalTrials.gov, NCT04427436, registered on 10 June 2020, https://clinicaltrials.gov/ct2/show/study/NCT04427436.

### Strengths and limitations of this study

► This study will be the first to use a multimodal measuring approach to determine neurophysiological responses of elderly with mild cognitive impairment and healthy controls.

► Differences between these two groups will be investigated with electroencephalography, functional near-infrared spectroscopy and heart rate variability at resting state and while performing cognitive tasks.

► This study will provide valuable information about certain neurophysiological parameters that are promising for an early identification of people who are at a higher risk of an overly age-related decline in cognitive performance.

► We hypothesise that this multimodal approach improves the classification accuracy between elderly with mild cognitive impairment and healthy controls as compared with a unimodal or a bimodal approach.

► Since this study is the first of its kind, it has an exploratory character with a relatively small sample size, but it should provide a basic concept for large-scale studies.

## INTRODUCTION

Mild cognitive impairment (MCI) is considered as a prodromal stage of (or transition phase to) dementia. The most common cause of dementia is Alzheimer's disease (AD).[1] It is estimated that approximately 50 million people are currently suffering from AD worldwide.[2] AD is a neurological disease with specific neuropathological (eg, amyloid-plaques, neurofibrillary tangles) and neurochemical features (eg, neurotransmitter deficits). On the behavioural level, AD is characterised by deficits of higher cortical functions such as memory, decision-making, visuospatial abilities, executive functioning and language.[3] Individuals who suffer from dementia require a high level of care and

support, and, thus, the disease entails a substantial burden for the healthcare systems.[4] For instance, in Germany, the annual costs are estimated at 42.6 billion Euros,[5] whereas the global costs are considered to have crossed the US$1 trillion threshold in 2018.[6] Given that the number of individuals with dementia will increase further as a consequence of the demographic change, these annual healthcare expenditures are expected to grow, too.

In contrast to patients with dementia, patients with MCI have only marginal limitations in activities of daily living but show some deteriorations in specific cognitive domains.[7] A person is considered to suffer from MCI if the following criteria are fulfilled: memory complaint, normal activities of daily functioning, normal general cognitive function, abnormal memory for age and not demented.[8] There are four types of MCI: amnestic MCI single domain, amnestic MCI multiple domain, nonamnestic MCI single domain and nonamnestic MCI multiple domain.[9] Patients with amnestic MCI show impairments in the performance of neuropsychological tests of episodic memory. Patients with nonamnestic MCI show impairments in cognitive domains other than memory, such as executive functions, language or visuospatial abilities.[8] For a correct detection of MCI, a careful and comprehensive neuropsychological test battery covering multiple cognitive domains is an important criterion.[8] Therefore, a correct detection of MCI by clinical data, regardless of whether single-domain and multiple-domain MCI are present, is relevant for our investigation. Hence, individuals with amnestic and nonamnestic MCI, based on a comprehensive clinical and neuropsychological assessment, without differentiating between single and multiple domain MCI, will be considered in our investigation. As these MCI are often not evident on a behavioural level, individuals with MCI are difficult to identify. During the stage of MCI, reduction of lifestyle-related risk factors, such as physical inactivity, may slow down or postpone the progression of this neurological syndrome.[10–12] So far, no drug has been approved for the symptomatic or causal treatment of MCI.[13] Nevertheless, an early diagnosis of MCI is indispensable for a better understanding of the neurobiological mechanisms mediating the transition of the inexorable age-related cognitive decline to dementia. This, in turn, may allow for a timely initiation of preventive strategies slowing, postponing or, at best, counteracting the transition to a serious manifestation of dementia.

The diagnosis of MCI based on clinical features alone is often challenging and relatively unreliable. Therefore, the identification of biomarkers that aid an early identification of neurodegenerative processes has become a focus of current research.[14] Neuroimaging methods such as MRI or positron emission tomography (PET) are popular tools in the investigation of neurodegenerative diseases. However, these methods have several disadvantages. They are expensive, have limited eligibility and the participant is not allowed to move. Additionally, the use of radioisotopes in PET results in exposure to ionising radiation, which involves detrimental health risks.[15 16] Biomarkers in the cerebrospinal fluid (CSF), such as Aβ42, constitute another approach for identifying AD pathology.[1] However, drawing CSF is stressful for the patient, can entail complications and does not necessarily yield unequivocal results. Thus, it is necessary to investigate other measurement methods able to detect a pathological cognitive decline timely while avoiding the above-mentioned limitations.

fNIRS is an optical neuroimaging technique that is based on the theory of neurovascular coupling and optical spectroscopy.[17] fNIRS allows the noninvasive investigation of cortical haemodynamic changes associated with brain activity.[18] In this regard, it is assumed that neuronal activity enhances the regional cerebral blood flow to the neuronal tissue at work which, in turn, leads to a local increase in the concentration of oxygenated haemoglobin (oxyHb) and decrease in deoxygenated haemoglobin (deoxyHb).[17] Since oxyHb and deoxyHb have different light absorption spectra, the relative activity-dependent concentration changes of oxyHb and deoxyHb can be determined and allow us to make conclusions about cortical brain activity.[16] More details on physiological and methodological background of fNIRS can be found in the referenced literature.[17 19 20]

The advantages of fNIRS are its portability, tolerance to motion artefacts (compared with EEG and MRI), higher temporal resolution (compared with MRI), low costs (compared with MRI and PET), it is easy to use and more participants are eligible than for MRI (eg, no restriction due to claustrophobia or metallic implants). The disadvantages are its lower spatial resolution (compared with MRI), low penetration depth, lack of standardisation in data analysis and, due to the delay in the haemodynamic response, there is a time delay in the signal curve, too.[16]

fNIRS has become a frequently used technique for the investigation of neurological diseases such as AD.[21–23] There is evidence in the literature showing that fNIRS offers a great potential to become a valuable tool to identify individuals with a higher risk of developing dementia as changes in cortical haemodynamic obtained during a standardised cognitive test allow to discern between healthy and diseased individuals.[24–31] Young participants showed a stronger activation in the left hemisphere in the more difficult task conditions. Elderly participants showed no lateralisation and a decreased activation in the difficult task condition. This phenomenon is called 'Hemispheric Asymmetric Reduction in OLD adults' model.[32]

Electroencephalogram (EEG) is another technique for quantifying neurophysiological processes. EEG measures electrical activity provoked by the firing of cortical neurons within the brain. EEG is an optimal tool for recording the complex dynamic neural activity due to the excellent temporal resolution in milliseconds and reasonable spatial resolution.[33] The potential to detect predementia AD/MCI condition has already been shown.[34–37]

In addition to the most commonly used method of frequency analysis, it is also recommended to apply other

more sophisticated EEG methods, like brain mapping, connectivity analysis or the analysis of event-related potentials (ERPs), which can lead to a further specification of the diagnosis of dementia.[38–42]

The specific neural activity in response to a certain stimulus can be measured by ERPs. The most common component we can find in response to sensory, cognitive or motor stimuli are P100, P200, P300, P600, N100 and N400, which are used in cognitive science.[43] Importantly, it is not sufficient to study the cortical modulation in isolation. Cortical functions are influenced by functional integration. EEG is an optimal technique to understand the communication between different areas of the brain by functional and effective connectivity analysis. Functional connectivity is defined as the correlation between different areas of the brain. Effective connectivity is defined as direct or indirect influence that one neural system exerts on the other neural system. The connectivity can be investigated by sources and EEG channels.[44–46] Beside connectivity, it is also possible to construct the brain map from the recording of electrical potentials with electrode distribution over the skull. The brain mapping can be helpful to identify activated structures of the cortex during task execution.[47 48]

ECG will be the third assessment method. In particular, we will determine the heart rate variability (HRV) from the ECG. The HRV describes the beat-to-beat variation of subsequent NN intervals. These fluctuations in the heart rate are regarded as an indicator of the functional state of the autonomic nervous system (ANS) and is related to psychophysiological aspects such as self-regulation on a cognitive, emotional, social and health level.[49 50] HRV is the result of the complex interaction between sympathetic and parasympathetic influences.[51 52] A relatively low HRV indicates an increased sympathetic state, an abnormal regulation and inadequate adaptation of the cardiovascular system and is a sign of depletion or pathology. Hence, HRV is a promising marker to diagnose pathological states[52] and an optimal HRV is critical to health and well-being.[53]

HRV measurements provide the advantages that they are easy to use, not expensive, portable, noninvasive, and the recordings are relatively easy to interpret. However, on the downside, HRV values are strongly influenced by individual differences, there is a relatively large number of influencing factors that have to be controlled for, and there is still no generally accepted consensus concerning data acquisition and data processing.[49 54 55]

Measures of HRV have been used to index vagal activity.[56] Root mean square of successive differences (RMSSD) between adjacent normal NN intervals, percentage of successive normal NN intervals differing more than 50 ms (pNN50) and power in high (0.15–0.40 Hz) frequencies (HF) are supposed to reflect vagally mediated HRV.[49 53 57] According to the neurovisceral integration model, neural structures responsible for affective, cognitive and physiological regulation are associated with vagally mediated cardiac function.[58] Especially the prefrontal cortex is associated with HRV measures as it is connected with the amygdala and cardiovascular system.[59] Vagally mediated HRV is supposed to be linked with 'a set of neural structures that have been implicated in cognitive, especially executive function'.[58] A positive connection between resting HRV and cognitive functioning has already been demonstrated.[60–64] In a recent meta-analysis, the effect size (ES) magnitude of HRV measures in the evaluation of autonomic dysfunction in older people with dementia was investigated.[65] Although the small ES does not support the use of HRV as a single biomarker to diagnose dementia, the results suggest autonomic dysfunction in dementia. Since biological processes such as ANS activity are complex and nonlinear, several authors suggest nonlinear HRV measures.[66–70] In this context, sample entropy recorded at rest, was related to a better cognitive performance, but traditional time or frequency indices were not.[69] The sensitivity and reliability of nonlinear measures such as Poincaré and detrended fluctuation analysis for mental effort tasks could be proved in healthy seniors.[67] In a recent study, D2, the RRI dimension correlation, could be better related to the mental workload than time or frequency indices.[66]

Cognition relies on complex neurophysiological processes. In particular, the solving of a cognitive task is associated with task-related changes in cerebral haemodynamics, cerebral electrical activity and changes in the ANS. These task-related neurophysiological changes can be assessed by fNIRS, EEG and HRV[71] separately (unimodal approach) or simultaneously (multimodal approach). A clear advantage of the multimodal assessment approach of task-related neurophysiological changes (eg, cognition-related brain activity) is its ability to reduce and/or to compensate for inherent limitations of a single measurement modality (eg, artefacts that are reflected in only one modality).[72] As single tests, they might be not accurate enough for differentiating healthy controls (HC) from cognitively impaired participants. EEG, fNIRS and HRV provide complementary information about different neurophysiological systems which, in turn, foster an improvement of the classification accuracy. Indeed, cognitive neuroscience research now focuses on the simultaneous acquisition by noninvasive modalities to improve their performance and information content.[15] Combined fNIRS-EEG,[73] EEG-HRV[74 75] and fNIRS-HRV[76] measurements were already applied in different fields of brain research. It was observed that classification accuracy can be improved when using different measurement modalities simultaneously.[77–84] However, so far and to the best of our knowledge, there is no published study available, which uses the above-described multimodal approach to investigate the neurophysiological cognition-related differences between healthy older individuals and older individuals with MCI.

The primary aim of this study is to investigate the difference between HC and cognitively impaired participants in neurophysiological signals. We will use fNIRS, EEG and ECG since they are noninvasive and provide some

advantages compared with MRI, PET and measures of CSF. As the most of the currently available studies used, to the best of our knowledge, only two different measurement modalities (mostly EEG and fNIRS), we hypothesise that our approach improves classification accuracy considerably. The second aim of this study is to investigate possible neurobehavioral relationships between measures of cognitive performance and measures of fNIRS, EEG and HRV. The third aim of this study is to systematically compare the classification accuracy of uni-, bi- and multimodal measurement approaches. By saying that, we wish to emphasise that this study has an explorative character aiming to identify cognition-related neurophysiological parameters that are promising for MCI detection. This study should provide a basic concept for further studies using a multimodal measurement approach and promote research for a better understanding of the neurobiological mechanisms leading to dementia.

## METHODS AND ANALYSIS
### Study design
This cross-sectional study will be conducted by a multidisciplinary team of researchers from Department of Sport Science at the Otto von Guericke University Magdeburg, the Medical Faculty of the Otto von Guericke University Magdeburg and the German Center for Neurodegenerative Diseases. The study protocol was approved by the Ethics Committee of the Otto von Guericke University Magdeburg (reference: 83/19) and is in accordance with the Declaration of Helsinki. This study was registered in ClinicalTrials.gov on the 10 June 2020.

### Participants
This study has an explorative character as there is, to the best of our knowledge, no comparable study available, which used measures from multiple modalities (eg, EEG, fNIRS, HRV) to differentiate between individuals suffering from MCI and HC. Thus, we considered studies comparing individuals with MCI and HC using, at least, one measurement modality[29 85 86] for our sample size estimation. In this context, the following calculations have been performed.

Based on the means of oxyHb of individuals with MCI and HC during a cognitive task in the study of Yang *et al*,[29] a sample size of 15 participants in each group will be needed to achieve a statistical power of 80%. Based on the HF nu values during standing position in one HRV study,[85] a sample size of 49 participants in each group will be needed to achieve a statistical power of 80%. Finally, based on the latency of P300 during an event-related task in one EEG study,[85] a sample size of 34 participants in each group will be needed to achieve a statistical power of 80%. All sample sizes were calculated a priori by using G*Power V.3.1.[87] Furthermore, studies that had applied EEG, fNIRS and ECG simultaneously in a cohort of healthy subjects[71 82 88] used sample sizes varying between 11 and 25 subjects. Given

the explorative character of this study and the intention to pave the way for future investigations with a larger sample size, we chose a relatively small sample with 30 MCI subjects and 30 HC constituting a tempered and conservative estimate to detect possible neurophysiological effects and/or trends.

This study will involve patients with MCI who have been diagnosed by an experienced neurologist, based on standardised clinical and neuropsychological criteria.[89] Healthy participants will be recruited by advertisements in local newspapers. Interested individuals will be informed about the aim of the study and first be screened by telephone to check for general eligibility according to our criteria. The two groups will be age, gender, handedness and education matched. At the beginning of the study, each participant will provide a personally signed and dated informed consent document indicating that the individual has been informed of all pertinent aspects of the study. Participants must be native German-speaking adults who are between 55 and 80 years old. All participants will be financially rewarded to compensate for their participation.

Exclusion criteria are:
► Other neurological diseases (ie, epilepsy, multiple sclerosis).
► Known severe cardiac diseases (ie, history of heart disease, severe cardiac insufficiency, heart failure, cardiac pacemaker, valvular defect, with or without stent implantation, heart attack).
► Stroke.
► Mental diseases (ie, schizophrenia, depression).
► Orthopaedic diseases (ie, bone fracture in last 6 months, symptomatic slipped disc).
► Muscular diseases (ie, myositis, tendovaginitis).
► Severe endocrinologic diseases (ie, manifest hypothyroidism or hyperthyroidism, adiposity (BMI >30), juvenile-onset diabetes).
► Injury or surgery in last 6 months.
► Use of illegal intoxicants or alcohol abuse (more than three times per week).
► Uncorrected poor eyesight or hearing.
► Colour blindness/red-green weakness.
► Pregnancy or breastfeeding.
► Using one of the following medications: betablocker, angiotensin-converting-enzyme inhibitor, antiarrhythmic drugs, neuroleptics, narcotic analgesics, benzodiazepines and psychoactive medications.

At the beginning of the study, all participants will be screened by using the CERAD (Consortium to establish a registry for Alzheimers's Disease) test battery.[90] This cognitive test battery evaluates the performance in semantic verbal fluency, word retrieval, constructional praxis, visual memory, verbal memory, global cognition (mini mental state examination) and motor speed (trail making test A). The CERAD has been shown to be a valid and reliable assessment tool to identify individuals with cognitive impairments.[91]

## Assessments

Participants will attend three visits within 1 week and undergo the following assessments:

1. Clinical assessment: CERAD and medical examination, taking blood sample (ie, Apolipoprotein 4 and brain-derived neurotrophic factor).
2. Sociodemographic assessment: questionnaires.
3. Neurophysiological and neuropsychological assessment: EEG/fNIRS/ECG simultaneously at resting state and during cognitive tasks.

## Sociodemographic assessment

Sociodemographic data, family history of AD/dementia and lifestyle factors of the participants will be obtained before the measurements via several questionnaires. We will record medication, together with measurements of height, weight, educational and physical activity level. The level of physical activity will be assessed via the questionnaire German-PAQ50+.[92] It will allow for this factor to be considered as a covariate. The health-related quality of life will be measured by the 36-Item Short Form Health Survey.[93] Sleep quality will be assessed via the Pittsburgh Sleep Quality Index.[94] Finally, the Food Frequency Questionnaire will be applied to assess the dietary habit of the participants.[95]

## Neurophysiological and neuropsychological assessment

Research staff collecting data are blinded concerning the cognitive status (MCI or HC) of the participant to avoid bias. EEG, fNIRS and ECG will be recorded simultaneously at resting state for 10 min. The measurement standards for the resting state measurement follow the recommendations of Laborde *et al.*[49] Accordingly, the participants are asked to sit on a comfortable chair with their knees bent at a 90° angle and their hands on their thighs. Furthermore, they are advised to relax, breath normally and move as little as possible. As recommended in the literature, the participants will rest in the above-described position for at least 5 min before the baseline recordings are obtained.[49] Subsequently, the participants complete three cognitive tasks while their neurophysiological signals are measured. The cognitive tasks are Stroop, N-back and a VFT. This procedure is comparable to the study of Yang *et al*[29] but in addition to this study, (1) we will conduct the assessment of EEG and ECG and (2) we will use modified versions of the cognitive tasks. Afterwards, a second resting-state measurement will be conducted (recovery phase).[49]

According to the multistage concept in psychophysiology of Fahrenberg, objective data of the cognitive performance, objective physiological data and subjective data of the personal feeling will be collected.[96] The latter will be recorded by the NASA-TLX questionnaire (National Aeronautics and Space Administration - Task Load Index). The NASA-TLX provides an overall workload score with six dimensions: mental demands, physical demands, temporal demands, own performance, effort and frustration.[97] The overall workload is the weighted average of these dimensions. After the cognitive tests, participants will be instructed to rate each dimension on a visual scale from 1 to 20 points. It is an easy to use tool and its results can be compared with participants' performance and their neurophysiological responses. The NASA-TLX is a widely used tool with a high reliability and validity.[98]

## Cognitive tasks

During the whole assessment, the participants are sitting in front of a computer screen on a comfortable chair and are asked to avoid head movements as much as possible. Before the experiment, the participants will be briefed on the task instructions and experimental design. The instructions are presented on a printed paper and explained verbally by the investigator. Each task block begins with a 5 s instruction cue that informs the participant about the task condition. All tasks will be administered via computer, which allows to measure reaction times and responses of the participants exactly. To synchronise all signals, temporal triggers are delivered simultaneously to EEG and fNIRS systems via the software Presentation (Neurobehavioral Systems, USA).

### *Stroop*

A computerised version of the Stroop test will be applied.[99] This widely used test requires the executive functions inhibition and cognitive control. These are crucial for completion of complex cognitive tasks and everyday activities.[100] The classic Stroop test elicits a conflict situation since the meaning of a colour-word and the ink colour do not match. Behavioural responses to these incongruent stimuli are usually slower and less accurate than responses to congruent stimuli. The behavioural difference between incongruent and congruent stimuli is called the Stroop effect, which has been used as an index of cognitive control.[101] Since the prefrontal cortex plays a predominant role in cognitive control[102] and has shown to be affected by MCI,[23] fNIRS optodes will be placed on the prefrontal cortex.

Our Stroop test includes three experimental conditions (pure congruent, pure incongruent and mixed congruent and incongruent stimuli) with three blocks in each condition (see figure 1). Prior to each block, a baseline measurement having nearly the same length as the task blocks will be applied in order to assess a baseline for the haemodynamic activity. During the baseline measurement, participants are requested to sit still and relax. According to the recent recommendations,[17] the duration of the baseline measurement should not be a multiple of 10 s to avoid the overlap with the Mayer waves. In each block, colour-words are consecutively presented in the middle of the computer screen. A black background was chosen to avoid overstraining the eyes. Each colour has a corresponding button. Four different colour-words will appear: 'RED', 'GREEN', 'BLUE' or 'YELLOW' in German language. In the congruent condition, the meaning of the colour-word and the ink colour

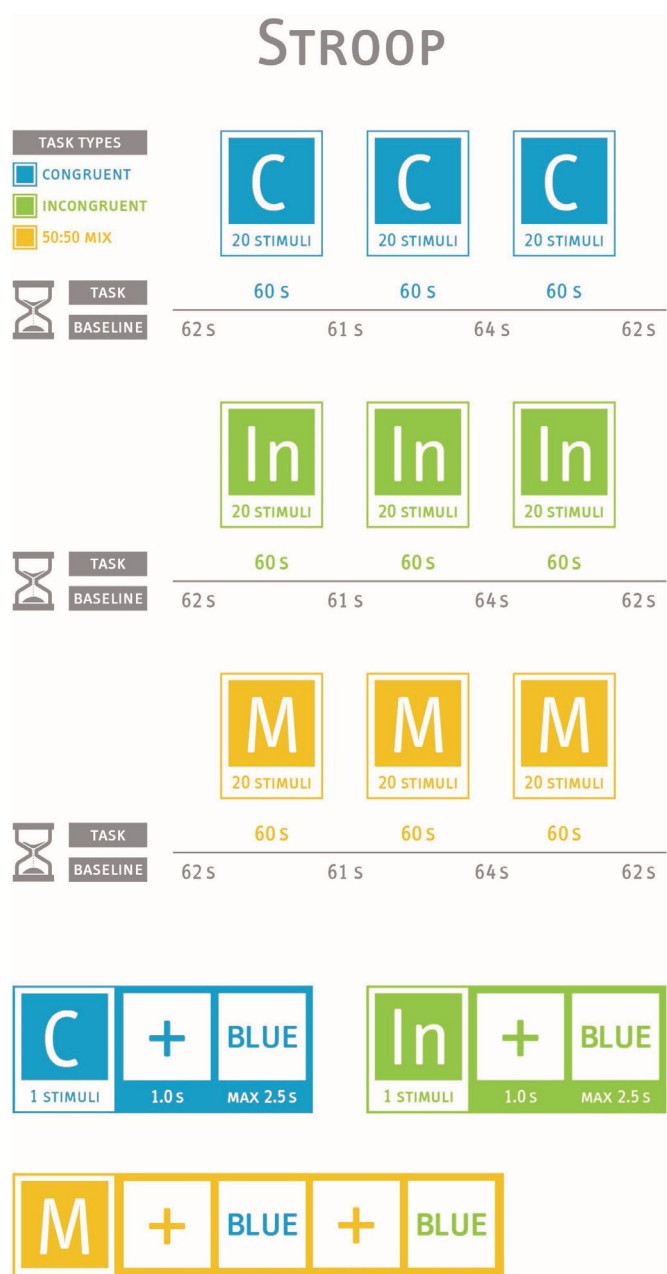

**Figure 1** Description of the Stroop paradigm. C, congruent condition; In, incongruent condition; M, mixed condition; s, seconds.

matches. In the incongruent condition, the colour-word is printed in an incongruent ink colour, for example, the word 'RED' is presented in blue colour. Participants are instructed to identify the colour of the word by pressing the appropriate button and not reacting on the meaning of the word. In this example, the participant must press the button for the ink colour (blue). Participants should react as fast and as correct as possible. In the third block, congruent and incongruent stimuli are presented. Participants are instructed to react on the colour of the word. Although the typical clinical version of the Stroop tests includes a pure block of neutral (congruent) stimuli and a pure block of incongruent stimuli, some authors

criticise this pure-block design since it has some limitations.[100] Therefore, a mixed block design with congruent stimuli within an incongruent condition demanding the cognitive function goal-maintenance capacities will be applied. The congruent stimuli within this condition promote the inappropriate but more automatically response of reading the word and produce larger Stroop effects than pure incongruent blocks.[100] This mixed design was already used in some studies.[103–107] Hence, we will use a pure incongruent and a mixed block. The mixed block consists of 50% congruent and 50% incongruent stimuli. The participants should react in both variants to the colour of the word. Prior to task, the participants will be adequately familiarised with the test by performing a sufficient number of practice trials.

### N-back

The N-back was first introduced by Kirchner[108] and is a frequently used task to measure working memory capacity. In our study, three conditions (0-back, 1-back and 2-back) will be used (see figure 2). Single-digit numbers with a presentation time of 1500 ms are presented consecutively in three blocks in the middle of the screen. The interstimulus interval is 500 ms. As in the Stroop test, a baseline measurement will be included prior to each task block and after the last block.

In the 0-back condition, the participants are asked to press the target button only when the number '7' appears. All other numbers must be ignored. In the 1-back condition, the participants are asked to press the target button only when two identical numbers appear in succession. In the 2-back condition, the participants are asked to press the target button only when the current number matches the second-last number displayed before. In all three conditions, 25% of stimuli are targets. As in the Stroop test, participants will be instructed to react as fast and as correct as possible. Prior to task, the participants will be adequately familiarised with the test by completing a sufficient number of practice trials.

### Verbal fluency test

Verbal fluency is a cognitive domain that worsens with the development of AD.[109] Therefore, we plan to use a VFT, which is based on the 'Regensburger Wortflüssigkeitstest'.[110] This test is frequently used in fNIRS studies investigating neurological diseases.[24 30 109 111] Based on these studies, our test consists of two conditions. Both conditions will be presented three times in a row for 30 s with a resting block for 31–34 s between each block. See figure 3 for a detailed description of the VFT paradigm. During the resting blocks, the participants are requested to sit still and avoid speaking. We use a phonological (letter) and a semantic (category) condition. The instructions will be displayed on a screen to avoid instruction bias. Participants are requested to avoid movements during this test.

In the phonological condition, the participants are instructed to pronounce as many German words

# N-Back

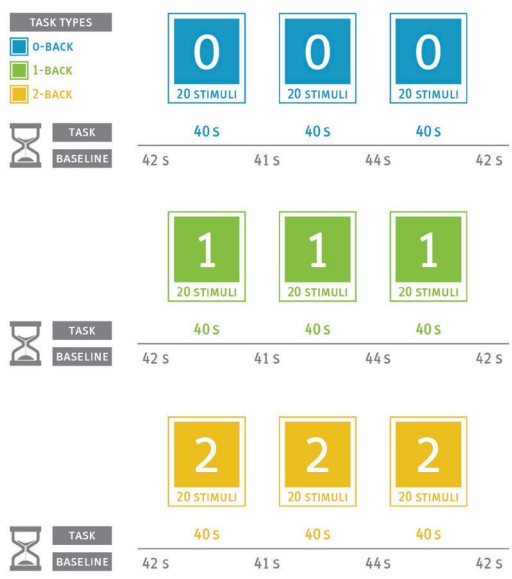

# Verbal Fluency Test

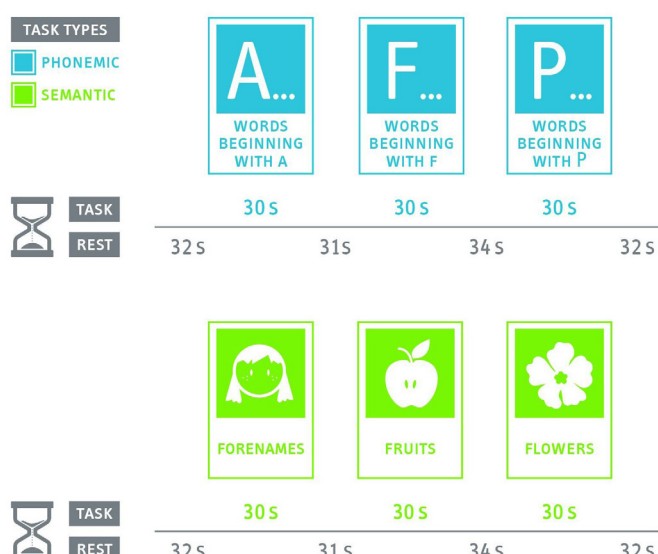

**Figure 3** Description of verbal fluency test paradigm. s, seconds.

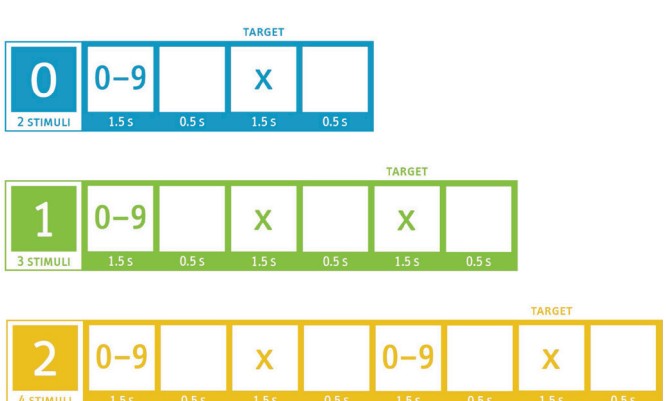

**Figure 2** Description of N-back paradigm. 0, 0-back; 1, 1-back; 2, 2-back; s, seconds.

(nouns, verbs, adjectives) as possible beginning with a specific letter ('A', 'F', 'P') without using proper names, numbers, repetitions or words in different forms or with different endings. In the semantic condition, participants are instructed to pronounce as many words as possible belonging to a specific category ('forenames', 'fruits', 'flowers').

## Neurophysiological assessment

### fNIRS

In this study, a portable continuous wave fNIRS system (NIRSport, NIRx Medical Technologies, Glen Head, New York, USA) will be used to record cortical haemodynamics with a frequency of 10.2 Hz. The fNIRS-system consists of the following: eight light sources which emit light at wavelengths of 760 and 850 nm, eight light detectors and a short-distance detector bundle (NIRx Medical Technologies, Glen Head, New York, USA), which allows to quantify changes in extracerebral layer (ie, blood flow in the scalp). As shown in figure 4, the fNIRS optodes will

be positioned according to the 10–20 EEG system[112] by using a standardised cap (EasyCap GmbH, Herrsching, Germany). We will perform a virtual and probabilistic spatial registration using the software fOLD (fNIRS Optodes' Location Decider)[113] and the Broadmann atlas[114] to assign the fNIRS measurement channels with long source-detector separation to specific brain regions (see online supplemental additional file 1 for detailed overview).

The data processing of fNIRS follows recent recommendations[17 115 116] and will be conducted using the latest version of the 'Homer' software package.[117] In brief, we will conduct the following processing steps: (1) exclude noisy channels by using enPruneChannels function, (2) convert raw light intensity changes into changes in optical density by using hmrIntensity2OD function, (3) perform motion artefact correction by using sophisticated filter methods (eg, wavlet filter—hmrMotionCorrectWavelet filtering function),[118] (4) perform a correction for physiological artefacts such as heart beat and instrumental noise by using hmrBandpassFilt function, (5) convert optical density data of both wavelengths via the modified Beer-Lambert law into concentration changes of oxyHb and deoxyHb by using the hmrOD2Conc function and an individually calculated differential path length factor,[119] (6) correct for extracerebral blood flow by using hmrDeconvHRRF_DriftSS function[120 121] and (7) perform a baseline correction and calculation of block averages for oxyHb and deoxyHb changes over all trials and for each measurement channel by using hmrBlockAvg function. In the final step, the preprocessed time series of oxyHb and deoxyHb are exported and the cognition-related

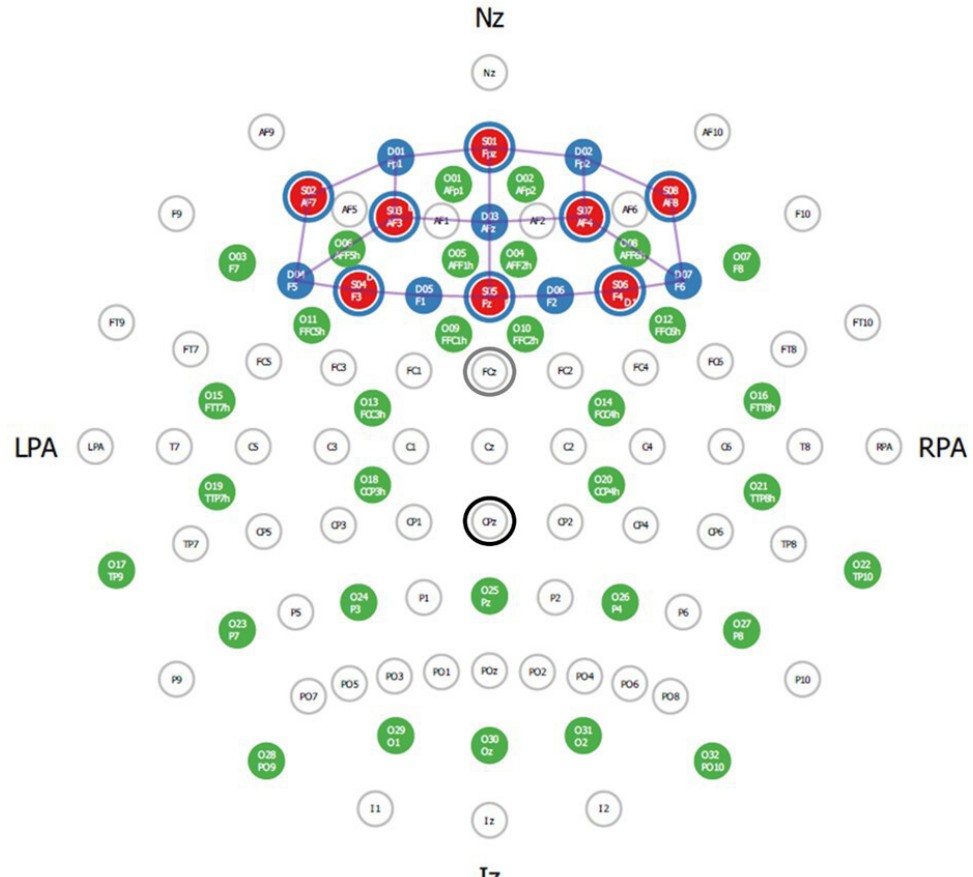

**Figure 4** Visualization of the positions of the EEG electrodes and fNIRS optodes. IZ, inion; LPA, left preauricular point; NZ, nasion; RPA, right preauricular point.

changes in these two chromophores are used for further statistical analysis.

*EEG*

EEG data will be acquired by the Brain Vision wireless MOVE system (Brain Products GmbH, Munich, Germany) along with Brain Vision Recorder V.1.21.0102 (Brain Products GmbH, Munich, Germany). The system will be triggered by the computer running the Presentation software. The EEG/fNIRS cap will be customised according to the 10–20 system with the provision of fNIRS optodes and 34 EEG channel slim-active electrodes (AFp1, AFp2, AFF5h, AFF1h, AFF2h, AFF6h, F7, F8, FFC5h, FFC1h, FFC2h, FFC6h, FTT7h, FCC3h, FCC4h, FTT8h, TTP7h, CCP3h, CCP4h, TTP8h, TP9, TP10, P7, P3, Pz, P4, P8, Po9, O1, Oz, PO10, Fz (reference) and ground).[80] The electrode location will be customised in the Brain Vision Recorder workspace according to the EEG elastic cap channel position in consideration of Theta and Phi values.[122] The impedance will be kept below 5 K before EEG data recording. The cognitive tasks will be recorded following a baseline measurement of 10 min. The EEG data will be recorded at 1.000 Hz. The outcome of the results will be focused on time and frequency domain of ERPs, power spectrum analysis, source localisation and connectivity.

The EEG recorded data will be analysed by the Brain Vision Analyzer V.2.2.0 (Brain Products GmbH, Munich, Germany). The EEG postprocessing will be divided into two blocks, whereas P100, P200, P300, P600, N100 and N400 ERPs,[2] power spectral density (PSD), connectivity and source localisation will be done for the N-back and Stroop, and PSD will be done for the VFT. The prerecorded EEG data will be resampled at 256 Hz and will be rereferenced to mean mastoids.[123] The referenced data will be filtered for ERPs high pass at 0.1 Hz and low pass at 40 Hz with 50 Hz notch filter. For PSD, connectivity and source localisation, a low-pass filter of 85 Hz will be used. The EEG data processing pipeline after rereferencing and filtering is shown in figure 5.

*ECG*

We will record the ECG of the participants during the experimental session with an ECG medilog AR12 plus (Schiller, Baar, Switzerland). This device consists of three channels with a sampling rate of 1.000 Hz. By using this device, we can record the NN intervals in millisecond range, which enables us to accurately calculate the HRV and to precisely identify cardiac arrhythmia. The raw ECG data will be uploaded to the Medilog Darwin Analysis Software package and analysed using the Kubios premium V.3.3 software package (University of Kuopio, Finland).

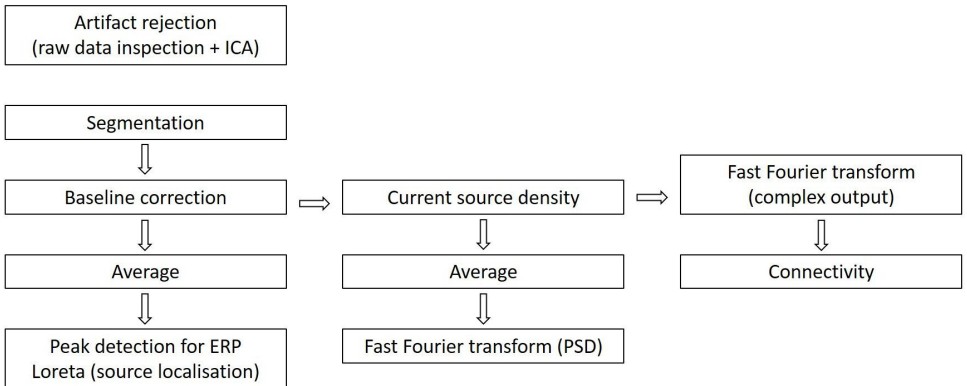

**Figure 5** EEG data processing pipeline. ICA, independent component analysis; Loreta, low-resolution electromagnetic tomography analysis.

The HRV analysis will focus on the following parameters: root mean square of successive differences beetween adjacent NN intervals (RMSSD), percentage of adjacent NN intervals differing more than 50 ms (pNN50) and power in HF (0.15–0.4 Hz). Further time and frequency domain parameters[124 125] will be analysed as well. In addition, we will also consider nonlinear HRV parameters to study the nonlinear dynamic properties that influence heart rate.[66] Due to the relatively long latency of the cardiac system,[53] the HRV (as the fNIRS parameters) will be averaged over each block of the three cognitive tasks. Following the recommendations of Laborde *et al*[49] we plan to record and analyse the HRV at three timepoints:

▶ At resting state before the cognitive tasks,
▶ During the cognitive tasks for each block.
▶ After the cognitive tasks at resting state in the recovery phase.

The baseline measurement before the cognitive tasks will serve as assessment of the vagal tone of the participants. The assessment of the vagal tone is important because 'vagally mediated HRV may serve to index the functional capacity of a set of brain structures that support the effective and efficient performance of cognitive executive function tasks including working memory and inhibitory control'.[58] The 'phasic HRV' will be evaluated through comparing the HRV at rest with the values recorded during the cognitive tasks.[49] It shows how the ANS reacts and how the participant adapts to cognitive demands. Finally, the HRV will be recorded after the cognitive tasks at resting state (recovery phase). This experimental design allows for investigation of tonic HRV for each of the three measurement points (baseline, during and after the cognitive tasks). Furthermore, we can assess the phasic HRV because we will be able to detect the change between baseline and event ('reactivity'), the change between task and post-event ('recovery') and the change between baseline and post-event.[49]

**Statistical analysis**

The current study is designed to investigate biomarkers that can identify participants with an increased risk for cognitive deterioration. Thereto, several biomarkers will be evaluated, as possible candidates, for an early identification of MCI. We are interested in the transition state between healthy ageing and MCI. Consequently, our statistical analysis will include:

▶ Analysis of the differences between HC and MCI with respect to cognitive performance and physiological parameters.
▶ Correlational analysis between cognitive performance and physiological parameters.
▶ Machine-learning-based logistic regression approach for classification into HC and MCI.

To investigate effects of group, t-test for normal distributed variables will be used, in other cases, the Mann-Whitney U test. Correction for multiple comparisons will be taken into account with false discovery rate.[126] To investigate the correlations, we will use Pearson or Spearman correlation analysis. To control for effects of age, gender, regular physical activity and education as confounders, partial correlation will be applied. Table 1 shows the measured physiological parameters of all three cognitive tasks. Concerning ECG analysis, we will also measure the HRV in resting state before (baseline) and after the cognitive tasks (recovery).

The main aim of our study is to develop EEG/fNIRS/HRV measures that discriminate among HC and MCI and show that this multimodal measuring approach is promising and accurate for identification of MCI. For that purpose, we will use and adapt the machine-learning algorithm already used for bimodal approaches.[77–79 81 82 84] In a first step, after preprocessing and artefact correction, parameters from the three modalities will be extracted and the most discriminating ones playing a role in the development of neurodegenerative diseases will be selected. Principal component analysis can help to reduce parameters to a manageable amount by removing components with the highest variance.[127] Following parameter extraction, multimodal combining algorithm will be used for classification. To classify the signals, support vector machine (SVM) will be applied. SVM is one of the most commonly used supervised classifiers in the field of pattern recognition and has been widely adopted in many brain signal studies.[72 77–79 81 128] SVM can define two or more

**Table 1** Physiological parameters of each device in all three cognitive tasks

| Cognitive task | Physiological measurement device | Main parameters |
| --- | --- | --- |
| Stroop (congruent and incongruent) | ECG | RMSSD, pNN50, HF (absolute and nu), non-linear parameters (SD1, SD2, ApEn, SampEn, DFA, D2, RPA, MSE) |
| | fNIRS | oxyHb, deoxyHb, totHb |
| | EEG | ERPs, PSD, connectivity and source localisation |
| N-back (0-, 1- and 2-back) | ECG | RMSSD, pNN50, HF (absolute and nu), non-linear parameters (SD1, SD2, ApEn, SampEn, DFA, D2, RPA, MSE) |
| | fNIRS | oxyHb, deoxyHb, totHb |
| | EEG | ERPs, PSD, connectivity and source localisation |
| VFT (semantic and phonological) | ECG | RMSSD, pNN50, HF (absolute and nu), non-linear parameters (SD1, SD2, ApEn, SampEn, DFA, D2, RPAn, MSE) |
| | fNIRS | oxyHb, deoxyHb, totHb |
| | EEG | PSD |

ApEn, approximate entropy; D2, correlation dimension; deoxyHb, deoxygenated haemoglobin; DFA, detrended fluctuation analysis; ERPs, event related potentials; HF, high frequency power in absolute and normalised units (nu) [0.15–0.4 Hz]; MSE, mutliscale entropy; oxyHb, oxygenated haemoglobin; pNN50, NN50 divided by the total number of NN intervals; PSD, power spectral density; RMSSD, root mean square of the successive differences between adjacent normal RR intervals; RPAn, recurrence plot analysis; SampEn, sample entropy; SD1, in Poincaré plot, the standard deviation perpendicular to the line-of-identity; SD2, in Poincaré plot, the standard deviation along the line-of-identity; totHb, total haemoglobin.

classes by constructing an optimal hyperplane maximising the margin of separation between the closest data points belonging to different classes. SVM can be used in linear as well as in non-linear classification scenarios based on the kernel trick.[129]

Another aim of the study is to compare the performance of unimodal, bimodal and multimodal systems. For that reason, classification should be performed separately using different kinds of parameter sets for comparison: EEG-only, fNIRS-only, HRV-only, bimodal (EEG +fNIRS, EEG +HRV, fNIRS +HRV) and a multimodal parameter set (EEG +fNIRS +HRV).

### Data management and safety

All participants will be assigned a code independently from their group allocation ensuring their anonymity. In order to guarantee the security of all data, the personal information and the data of this study will be collected and handled exclusively by the involved researchers. All data and participants' information will be handled according to the institutional data management policy of the University of Magdeburg. The original documents will be kept by the main investigators of the study at the Otto von Guericke University Magdeburg and the German Center for Neurodegenerative Diseases.

### Patient and public involvement

Subjects will be involved in the study as we will evaluate their acceptability for our measurement procedure. We will collect feedback on the study procedure from the participants. We will continue to work with stakeholders including the medical faculty and companies working in the field of neurodegeneration and dementia.

### SUMMARY

Due to the demographic change and the increase in the individual life expectancy, the worldwide economic health-care costs to treat individuals suffering from dementia will increase considerably. Currently, there is no treatment available, which would allow to heal this neurological disease. Consequently, researchers now focus more on an early diagnosis of preclinical stages of AD (eg, MCI) in order to initiate preventive actions timely. In this regard, neurophysiological signals could be promising biomarkers of such preclinical AD stages because they are, unlike behavioural performance, deemed to be less affected by learning or practice effects and provide insights into possible compensatory processes being not readily observable at the behavioural level. Hence, physiological signals can be helpful to detect changes in cognitive performance more precisely.[130] In this context, it is assumed that the investigation of new neurophysiological biomarkers, which allows to identify MCI more easily and accurately, are necessary to better understand and monitor the disease progression.[11]

In line with the previous mentioned assumptions, the aim of this study is threefold. At first, this investigation aims to determine the differences in neurophysiological responses of HC and MCI participants. The second aim is to investigate possible neurobehavioral relationships between measures of cognitive performance and neurophysiological responses. The third aim is to elucidate whether a multimodal measurement approach can help to identify individuals with MCI more accurately and reliable than a unimodal or bimodal approach. Thereto, three complementary measuring modalities, namely, fNIRS, EEG and ECG/HRV, will be used simultaneously to assess different neurophysiological responses, which are associated with cognitive processes. To the best of our knowledge, this is the first study using these three modalities simultaneously in patients with MCI and cognitive HC. Given the explorative character of our study, our sample consists of 30 healthy controls (HC) and 30 patients with MCI whose neurophysiological signals will be recorded at a resting state and while performing three established cognitive tasks (Stroop, N-back and a VFT). We are aiming to detect certain neurophysiological parameters that are promising for an early identification of people who are at a higher risk of an overly age-related decline in cognitive performance (ie, MCI detection).

In this regard, we hypothesise that the multimodal approach improves the classification accuracy between

patients with HC and MCI as compared with a unimodal or a bimodal approach. If our hypothesis is verified, this study will pave the way for further research on multi-modal measurement approaches for dementia research. Such upcoming research will use a larger sample size to examine the noninvasive biomarkers characterising an early detection of nonphysiological decline in cognitive performance in more detail.

## ETHICS AND DISSEMINATION

Ethics approval was obtained for the study from the Ethics Committee of the Otto von Guericke University (reference: 83/19), and informed consent for participation will be obtained from all participants. Data will be shared with the scientific community no more than 1 year following completion of study and data assembly.

**Contributors** BG, FH, MD and TAG designed the study protocol. AH and NGM conceptualised and secured funding for the research study. BG, FH and MD contributed to the application to the local Ethics Committee. BG wrote the first draft of this publication with contributions from FH (fNIRS) and TAG (EEG). FH, MD, SD, NGM, IB and AH revised the initial draft. FH, TAG and SD settled the trial materials. IB, NGM and AH were responsible for the final study protocol. BG, FH, MD and TAG will participate in data collection. All authors read and approved the final manuscript.

**Funding** This work is supported by the joint project 'MyFit' funded by the European Regional Development Fund (grant number: ZS/2018/08/94206). The funder has no role in the design or conduct of the study.

**Competing interests** None declared.

**Patient consent for publication** Not required.

**Provenance and peer review** Not commissioned; externally peer reviewed.

**ORCID iD**
Bernhard Grässler http://orcid.org/0000-0003-0434-7194

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
