## [Reviewer comments · BMJ Open]

ARTICLE DETAILS

TITLE (PROVISIONAL)	Multimodal measurement approach to identify individuals with mild cognitive impairment: study protocol for a cross-sectional trial
AUTHORS	Grässler, Bernhard; Herold, Fabian; Dordevic, Milos; Gujar, Tariq; Darius, Sabine; Böckelmann, Irina; Müller, Notger; Hökelmann, Anita

VERSION 1 – REVIEW

REVIEWER	R Finlayson Vanderbilt University Medical Center, Psychiatry
REVIEW RETURNED	09-Jan-2021

GENERAL COMMENTS	This is an important project. Not only might it be useful to distinguish dementia from MCI, but it could also improve the ability to screen older people in safety-sensitive occupations for deterioration in cognitive functioning. I would be interested in how sensitive these tests would be in highly educated professionals. I would also suggest that measures of physical fitness will be an important factor.
--

REVIEWER	Lucette Cysique University of New South Wales
REVIEW RETURNED	29-Jan-2021

GENERAL COMMENTS	The manuscript presents a very well written study protocol for a cross-sectional study comparing 20 people with formally diagnosed MCI and 20 demographically matched controls on multimodal neurophysiological measures: functional near-infrared spectroscopy (fNIRS), electroencephalography (EEG) and heart rate variability (HRV) via electrocardiography (ECG). The study is presented as a proof of concept and is certainly innovative. The introduction convincingly presents the need for this new investigation. The methods and analyses are very well described and easy to follow. I have a few queries: 1. In the participants section, more details about the study on which the sample size is based are needed and this study was only using fNIRS-study with the N-back task. Should the authors consider other studies with the other methods they will include to have a more balanced approach for their sample size estimation?2. The authors state "This study will involve MCI patients who have been diagnosed by an experienced neurologist, based on standardized clinical, neuropsychological and imaging criteria". The reader would need to know exactly what type of "imaging" the authors are referring to and what implications it may have for the
--

	current study. 3. Some criteria of exclusion need greater refinements including other neurological diseases, severe cardiac diseases, mental diseases. Will stable and well treated people for moderate versions of these conditions be systematically included or excluded? especially cardiac? Will any use of illegal intoxicants or alcohol abuse be excluded or only a harmful level? 4. I am wondering if the planned inclusion of the wide age range 55-80 may lead to variance in the data that could be due to late onset versus late onset dementia emergence and that the planned sample size will not be able to handle this key variability. I am actually concerned overall that the sample size is way too small. 5. Other critical missing information are the genetic risk for dementia (APOE4 and whether individual may have early genetic form of dementia in their family) 6. Finally, the sample size will not be able to deal with the potential variable profile of MCI that the authors have delineated in their introduction. Could the authors also comment on this?
--	--

VERSION 1 – AUTHOR RESPONSE

Comments Reviewer	Comments Authors	References
Dr. R Finlayson, Vanderbilt University Medical Center Comments to the Author: This is an important project. Not only might it be useful to distinguish dementia from MCI, but it could also improve the ability to screen older people in safety-sensitive occupations for deterioration in cognitive functioning. I would be interested in how sensitive these tests would be in highly educated professionals. I would also suggest that measures of physical fitness will be an important factor.	Thank you very much for your valuable and constructive comments. We fully agree with your opinion that education is an important factor influencing cognitive performance in older individuals. To account for the influence of education, we will quantify the educational level of the subjects using a standardized questionnaire (as stated in line 334 in our manuscript). By doing so, we can control for education in our statistical analysis. We will also investigate sensitivity of the tests on both low and highly educated individuals. We totally agree with your opinion that physical fitness is an important lifestyle factor and contributes to the preservation of mental health as well as cognitive performance. Given that physical fitness has several subdomains (e.g., cardiorespiratory fitness, muscular fitness) and do not necessarily include all physical movements conducted (e.g.,	Budde, H., Schwarz, R., Velasques, B., Ribeiro, P., Holzweg, M., Machado, S., . . . Wegner, M. (2016). The need for differentiating between exercise, physical activity, and training. Autoimmunity Reviews, 15(1), 110–111. https://doi.org/10.1016/j.autrev.2015.09.004 Caspersen, C. J., Powell, K. E., & Christenson, G. M. (1985). Physical activity, exercise, and physical fitness: Definitions and distinctions for health-related research. Public Health Reports, 100(2), 126–131.

	leisure time activities such as dancing, take stairs to get to your flat), we quantify the regular physical activity level via a standardized questionnaire (GPAQ 50+). In other words, we quantify the regular physical activity level rather than a single domain of physical fitness because the term “physical activity” comprises all bodily movements which lead to a certain energy expenditure and thus covers all physical activities which are necessary to achieve a specific level of physical fitness (e.g., regular endurance training) (Budde et al., 2016; Caspersen et al., 1985). Hence, the construct “physical activity” is more comprehensive than a single physical fitness domain. In addition, we will consider the regular physical activity level as a covariate in our analysis (see line 530 in the revised manuscript).	
In the participants section, more details about the study on which the sample size is based are needed and this study was only using fNIRS-study with the N-back task. Should the authors consider other studies with the other methods they will include to have a more balanced approach for their sample size estimation?	Thank you very much for your suggestion. We have now taken this into account and calculated the required sample size based on two further studies (please see chapter 2.2. in the manuscript): 1) EEG study from Lai et al., 2010, whereby we used the latency of the event-related potential P300 of MCI subjects and healthy controls (HC) and 2) HRV study (Nicolini et al., 2014), from which we used the HF nu parameter. These two indices are very important parameters for our analysis. Based on these studies, we calculated a sample size of 34 and 49 subjects per group, respectively. Based on the study of Niu et al. (2013), 15 subjects per group are required. Therefore, we decided to increase the sample size from 20 to 30 subjects per group in our study, to increase the power. Additionally, there are no comparable studies combining all three measurement methods to differentiate between individuals with MCI and HC available up to date, which	Lai, C.-L., Lin, R.-T., Liou, L.-M., & Liu, C.-K. (2010). The role of event-related potentials in cognitive decline in Alzheimer's disease. Clinical Neurophysiology : Official Journal of the International Federation of Clinical Neurophysiology, 121(2), 194–199. https://doi.org/10.1016/j.clinph.2009.11.001 Nicolini, P., Ciulla, M. M., Malfatto, G., Abbate, C., Mari, D., Rossi, P. D., . . . Lombardi, F. (2014). Autonomic dysfunction in mild cognitive impairment: Evidence from power spectral analysis of heart rate variability in a cross-sectional case-control study. PLoS ONE, 9(5). https://doi.org/10.1371/journal.pone.0096656

	prevents us to directly compare our parameters. Therefore, the a-priori calculation of the required sample size is relatively difficult. As written above, our calculation is now based on these three studies that used at least one of our measurement methods to compare individuals suffering from MCI and HC. We would also like to stress that our study has a novel character and aims to provide the basic concept for upcoming studies with larger sample size.	
The authors state "This study will involve MCI patients who have been diagnosed by an experienced neurologist, based on standardized clinical, neuropsychological and imaging criteria". The reader would need to know exactly what type of "imaging" the authors are referring to and what implications it may have for the current study.	The primary criterion for group allocation is the score of the Mini Mental State Examination (MMSE). Individuals with a score below 27 are further screened by an experienced neurologist (NGM) which provides the final diagnosis. Hence, our group assignment is based on an objective measure (MMSE score) which is further verified by a clinical neurologist. This procedure has been established in the literature (O`Bryant et al., 2008). In addition, we thank the reviewer for his hint towards "imaging" and excuse us for this misleading sentence. The sentence as revised as follows: "This study will involve MCI patients who have been diagnosed by an experienced neurologist, based on standardized clinical and neuropsychological criteria".	O`Bryant, S. E., Humphreys, J. D., Smith, G. E., Ivnik, R. J., Graff-Radford, N. R., Petersen, R. C., & Lucas, J. A. (2008). Detecting dementia with the mini-mental state examination in highly educated individuals. Archives of Neurology, 65(7), 963–967. https://doi.org/10.1001/archneur.65.7.963

Some criteria of exclusion need greater refinements including other neurological diseases, severe cardiac diseases, mental diseases. Will stable and well treated people for moderate versions of these conditions be systematically included or excluded? especially cardiac? Will any use of illegal intoxicants or alcohol abuse be excluded or only a harmful level?	Thank you for pointing out this shortcoming. We have considered your suggestions and further specified our exclusion criteria. In this context, we add to our manuscript that (i) individuals who consume illegal intoxicants or drink alcohol more than three times per week will be excluded., (ii) that individuals with a history of heart disease will be excluded, (iii) that individuals taking medications that have been shown to affect HRV will be excluded. Of course, it is difficult to recruit older individuals who are not taking any medications at all and have no medical conditions, even minor ones as this does not reflect the overall population and hamper the generalizability of our findings. To account for the possible influence of medications, all participants were asked to list the medications they are currently using. The exclusion criteria read in the revised version of the manuscript as follows:  • neurological diseases (i.e., epilepsy, multiple sclerosis) • known severe cardiac diseases (i.e., history of heart disease, severe cardiac insufficiency, heart failure, cardiac pacemaker, valvular defect, with or without stent implantation, heart attack) • stroke • mental diseases (i.e., schizophrenia, depression) • orthopedic diseases (i.e., bone fracture in last six months, symptomatic slipped disc) • muscular diseases (i.e., myositis, tendovaginitis) • severe endocrinologic diseases (i.e., manifest hypothyroidism or hyperthyroidism, adiposity [BMI >30], juvenile-onset diabetes) • injury or surgery in last six months • use of illegal intoxicants or alcohol abuse (more than three times per week week) 	
---	---	--

	 • uncorrected poor eyesight or hearing • color blindness / red-green weakness • pregnancy or breastfeeding • using one of the following medications: betablocker, ACE inhibitor, antiarrhythmic drugs, neuroleptics, narcotic analgesics, benzodiazepines and psychoactive medications 	
I am wondering if the planned inclusion of the wide age range 55-80 may lead to variance in the data that could be due to late onset versus late onset dementia emergence and that the planned sample size will not be able to handle this key variability. I am actually concerned overall that the sample size is way too small.	We thank the reviewer for her/his constructive feedback. We refer the reviewer to his first concern in which we addressed the issue of the sample size in sufficient detail. We decided to increase the sample size from 20 to 30 subjects per group. We believe that this sample size justifies the age range of 55-80 years. Additionally, this age range allows us to recruit the required sample size, and it reflects a large group of the overall population in which cognitive deficits (e.g., MCI) arise. Nevertheless, it was also our intention to introduce age-related variance in the data, so we could obtain more generalizable results. Still, age will be included as a covariate in the statistical analysis. Finally, with regards to one of the primary goals of the study, age	

	distribution will be the same for both groups to eliminate the influence of age on possible group differences.	
Other critical missing information are the genetic risk for dementia (APOE4 and whether individual may have early genetic form of dementia in their family)	Thank you for this important suggestion. One of our questionnaires asks the participants for their familial history of dementia. We agree with the reviewer that APOE4 is an important genetic factor increasing the risk of dementia and is known to influence functional brain activity. To take this concern into account, we will assess APOE4. To reflect this idea, we have added the assessment of APOE4 in the manuscript (please see line 325/326). Given that APOE4 is well known as a risk factor for Alzheimer's Disease (O'Donoghue et al 2018), we will measure levels of APOE4 in all our study participants also.	O'Donoghue, M. C., Murphy, S. E., Zamboni, G., Nobre, A. C., & Mackay, C. E. (2018). Apoe genotype and cognition in healthy individuals at risk of Alzheimer's disease: A review. Cortex; a Journal Devoted to the Study of the Nervous System and Behavior, 104, 103–123. https://doi.org/10.1016/j.cortex.2018.03.025
Finally, the sample size will not be able to deal with the potential variable profile of MCI that the authors have delineated in their introduction. Could the authors also comment on this?	We are thankful for this thoughtful comment. We have included a description of the different MCI subtypes in our introduction because we feel that such important background information should be provided to the reader. Based on our a-priori defined inclusion and exclusion criteria, the individuals suffering from MCI and HC should not differ in the most important factors (age, gender, education, regular physical activity level) and should, in turn, yield a homogeneous sample. Most importantly, based on the careful and comprehensive clinical and neuropsychological assessment of the potential participants, we only consider individuals with amnesic and non-amnesic MCI without differentiating between single and multiple domain MCI. To reflect this change, we have	

added the following sentences to our manuscript (see line 114-124):]. Patients with amnesic MCI show impairments in the performance of neuropsychological tests of episodic memory. Patients with non-amnesic MCI show impairments in cognitive domains other than memory, such as executive functions, language or visuospatial abilities[8]. For a correct detection of MCI, a careful and comprehensive neuropsychological test battery covering multiple cognitive domains is an important criterion[8]. Therefore, a correct detection of MCI by clinical data, regardless of whether single domain and multiple domain MCI are present, is relevant for our investigation. Hence, individuals with amnesic and non-amnesic MCI, based on a comprehensive clinical and neuropsychological assessment, without differentiating between single and multiple domain MCI, will be considered in our investigation.

VERSION 2 – REVIEW

REVIEWER	Lucette Cysique University of New South Wales
REVIEW RETURNED	12-Apr-2021
GENERAL COMMENTS	The authors have addressed my comments